# Features and Extra-Striate Body Area Representations of Diagnostic Body Parts in Anger and Fear Perception

**DOI:** 10.3390/brainsci12040466

**Published:** 2022-03-31

**Authors:** Jie Ren, Rui Ding, Shuaixia Li, Mingming Zhang, Dongtao Wei, Chunliang Feng, Pengfei Xu, Wenbo Luo

**Affiliations:** 1Research Center of Brain and Cognitive Neuroscience, Liaoning Normal University, Dalian 116029, China; rejeaff@gmail.com (J.R.); psychray1989@gmail.com (R.D.); lsx91@lnnu.edu.cn (S.L.); zmm1001psy@lnnu.edu.cn (M.Z.); 2Key Laboratory of Brain and Cognitive Neuroscience, Dalian 116029, China; 3Faculty of Psychology, Southwest University, Chongqing 400715, China; dongtao@swu.edu.cn; 4School of Psychology, South China Normal University, Guangzhou 510631, China; chunliang.feng@m.scnu.edu.cn; 5Faculty of Psychology, Beijing Normal University, Beijing 100875, China; pxu@bnu.edu.cn

**Keywords:** bodily perception, Bubbles paradigm, diagnostic body parts, fMRI, EBA

## Abstract

Social species perceive emotion via extracting diagnostic features of body movements. Although extensive studies have contributed to knowledge on how the entire body is used as context for decoding bodily expression, we know little about whether specific body parts (e.g., arms and legs) transmit enough information for body understanding. In this study, we performed behavioral experiments using the Bubbles paradigm on static body images to directly explore diagnostic body parts for categorizing angry, fearful and neutral expressions. Results showed that subjects recognized emotional bodies through diagnostic features from the torso with arms. We then conducted a follow-up functional magnetic resonance imaging (fMRI) experiment on body part images to examine whether diagnostic parts modulated body-related brain activity and corresponding neural representations. We found greater activations of the extra-striate body area (EBA) in response to both anger and fear than neutral for the torso and arms. Representational similarity analysis showed that neural patterns of the EBA distinguished different bodily expressions. Furthermore, the torso with arms and whole body had higher similarities in EBA representations relative to the legs and whole body, and to the head and whole body. Taken together, these results indicate that diagnostic body parts (i.e., torso with arms) can communicate bodily expression in a detectable manner.

## 1. Introduction

Humans are able to detect and identify bodily expressions, which is crucial for social interaction. Bodily expressions are major sources of social and emotional information beyond facial expressions [1,2,3], especially when communicating at a considerable distance [4]. For example, a happy body is associated with large and broad motion patterns [5,6]. However, it remains largely unclear regarding what, in terms of diagnostic body parts, is indispensable for decoding bodily expressions, and what underlying neural mechanisms accompany the process. Diagnostic body parts are critical to effective and efficient social communication. This study used the Bubbles paradigm to uncover the diagnostic body parts and functional magnetic resonance imaging (fMRI) methods to investigate how they were decoded in body-selective cortical regions.

Previous studies have demonstrated that body parts have distinct biological motion patterns that modulate bodily expression perception. As long ago as the 19th century, Darwin [7] proposed that head motion indicates large amount of emotional information, through tilt direction: an upward orientation conveys joy, and a downward orientation conveys shame or sadness. This conjecture was later supported by evidence showing that people perceived emotion corresponding to specific head motions [8,9,10]. Even simple leg and foot gestures exert influence on the evaluation of people’s attitude [11]. These studies showed that separate body parts play important roles in forming bodily expression perception.

Perception of bodily expressions depends on how motion patterns are processed by a balance between encoding and decoding [10,12,13]. Encoding refers to expressing one’s inner emotion with nonverbal information. For example, slower and fewer overall body movements can express an unhappy or sad state in dance [14], though these movements are usually interpreted as neutral expressions in daily life. Decoding refers to extracting the nonverbal cues to interpret bodily expression. When an observer had difficulties identifying bodily expression from an ambiguous face, they would automatically extract information from other body parts [15,16]. Studies employing whole-body stimuli have found that decoding bodily expression usually involves how to decode body parts such as arm or hand gestures [17,18]. Specifically, anger is often recognized through arm movement [19], clenched or shaking fists and hitting gestures [20]. Fear is usually accompanied by defensive reactions [10,21], such as placing arms over the chest [16,20], or covering the face with hands [18]. Neutral has more expanded torso and limb shapes to express relaxed states [22]. However, these studies failed to clearly explain: (1) whether those body parts were all diagnostic for decoding bodily expression; (2) other body parts, such as the torso, containing the information of leaning forward or backward, were also helpful for behavior identification.

Moreover, emotions drive bodily responses [23], because emotions evoke specific patterns of autonomous nervous system activity [24], which further lead to discrete ‘feeling fingerprints’ of the human body [25,26]. In these works, many types of emotion showed that torso and arms were significantly associated with bodily sensations, such as anger and fear. The evidence from body feelings of emotion possibly explained how people perceive bodily expressions. We therefore speculated that perceiving anger or fear was probably achieved by qualifying whether the torso, arms and hands are contracted or expanded. We combined torso, arms and hands into an integrated part, and hypothesized that diagnostic features for perceiving fear and anger consist of the combined diagnostic body parts: torso with arms.

We adopted the Bubbles paradigm to identify diagnostic features for decoding static bodily expressions. The paradigm applied the theory of reverse correlation to model the mental representations by reverse correlating all the possible information projected onto the retina, with the observer’s corresponding response [13,27]. For instance, one observer selected some specific faces as happy expressions. Researchers computed significantly overlapped regions of his/her selected faces as diagnostic features. As the human visual system can be sampled into five orthogonal spatial frequency bands responsible for transmitting different scales of information, researchers always compute the diagnostic information independently for each frequency band. The Bubbles paradigm has been used for identifying human faces [27] and scenes [28]. For faces, many experiments have been designed to identify gender [27,29], identity [29], expression [27,29] and age [30]. In recent years, Jack et al. promoted reverse correlation when studying dynamic facial expressions [31,32], and they revealed cultural differences in features for perceiving dynamic faces between the East and the West [33]. To our knowledge, few studies have employed the Bubbles paradigm for bodily expression perception in human participants. By analogy with facial expressions, we intended to apply this method to how bodily expressions were decoded through diagnostic body parts.

Then, our next purpose was to reveal the influence of diagnostic body parts on the perception of bodily expressions. The distinction between body form and body action should be noticed, because they are intimately linked to body parts and bodily expressions. The evidence from brain lesions demonstrated that body form and body action can be double dissociated, indicating the functions were from different brain areas or structures [34]. Body action recognition can be associated with motor-related cortices, such as the somatosensory cortex, supplementary motor area and ventral premotor cortex [35,36]. Perception of body form involves independent neural structures, such as the fusiform body area (FBA), extra-striate body area (EBA) and posterior superior temporal sulcus [37]. In primates, the neurons in the temporal cortex drive the responses to human body parts [38]. In humans, the EBA and FBA, both of which are located near the occipito-temporal regions [39], have been identified to be preferential in processing body-related stimuli [40,41]. Particularly, the FBA is sensitive to whole-body stimuli, while the EBA is sensitive to images of neutral bodies and body parts [37,42]. The EBA also contains neural populations overlapped for form and action perception [34]. In other words, the EBA may drive the perception of body motion expressed by body parts. However, the EBA representation of body parts (i.e., arms and hands) remains elusive which is diagnostic for the perception of bodily expressions. We therefore predict that diagnostic features for perceiving body parts would engage the involvement of the EBA: (1) the representations of a torso with arms would be more correlated with the representations of whole bodies than that of other body parts; (2) the representations of angry body parts would be more correlated with the representations of fearful body parts than that of neutral body parts. This was motivated by the idea that the EBA possibly drives the abstraction of diagnostic features from the torso with arms. Combined with representational similarity analysis (RSA) [43], we conducted a follow-up fMRI experiment to reveal the influence of diagnostic body parts for the perception of bodily expression.

In addition, the limbic structures are sensitive to the arousal of negative visual stimuli. The brain structures of the caudate, thalamus and anterior insula are core hubs of the salience network, involved in the function of experiencing negative emotion [44,45]. The amygdala of the limbic structures plays important roles in the rapid automatic perception of body emotion, such as fear [46]. There are fine-grained distinctions of related sub-cortices between anger and fear perception [23,45,47,48]. The neural function for fear perception is characterized by fast and automatic processing, while anger perception depends on consciousness to some extent. In the present study, we expected that subcortical activations would be involved in the perception of emotional body parts.

Together, this study comprises two experiments to reveal diagnostic body features in bodily expression categorization and to uncover the corresponding neural patterns. In the behavioral experiment, participants viewed a variety of incomplete body images and indicated the emotion they perceived (i.e., anger, fear and neutral). A separate group of participants were tested in the fMRI experiment, where they were asked to identify different body parts. Our findings may provide converging evidence to support how our brain decodes the diagnostic features.

## 2. Materials and Methods

### 2.1. Participants

In the Bubbles experiment, 33 college students (18 females, mean age ± s.d. = 22.39 ± 2.83 years) were recruited from Liaoning Normal University via advertisement. In the fMRI experiment, 24 students (14 females, mean age ± s.d. = 21.48 ± 2.01 years) were recruited. All of the participants had normal or corrected-to-normal vision and were right-handed. The experiments were approved by the local ethics committee. Written informed consent was provided before participation. No neurological or psychiatric history was reported. After the experiments, participants were financially compensated.

### 2.2. Stimuli Presentation

In the Bubbles experiment, 12 body images of 4 different actors (2 female) expressing 3 bodily expressions (anger, fear and neutral) were selected from the BEAST stimulus set [49]. The mean (s.d.) of the vertical and horizontal extent of bodies was 293 (6.7) and 98 (14.9) pixels. They were therefore imbedded in a grayscale background (grayscale = 128) with the identical size of 310 × 245 pixels, presented at 6.90° × 5.47° of visual angle on a computer screen with a resolution of 1024 × 768 pixels and a refresh rate at 60 Hz. To sample the body features in different spatial scales, we decomposed one image into five nonoverlapping spatial frequency (SF) bandwidths of one octave each, with cut-offs at 123 (22.4), 61 (11.2), 31 (5.6), 15 (2.8), 8 (1.4) and 4 (0.7) cycles/image (c/deg of visual angle), respectively. The decomposition was processed by the toolbox of *matlabPyrTools* for MATLAB (https://github.com/gregfreeman/matlabPyrTools; accessed on 10 April 2012). The size of each octave was defined based on the image size. For example, the highest SF band expressed one cycle by 2 × 2 pixels. This SF layer was represented by 310/2 (in the vertical direction) and 245/2 (in the horizontal direction) cycles/image (cpi) [12,50]. We used horizontal SFs of 123 cpi here to measure the high-SF octave according to previous studies [51,52]. This process of peeling off each SF layer was applied recursively. Each band was independently sampled with a number of randomly positioned Gaussian bubbles (windows) to generate a bubbles mask (each in the second row of Figure 1). The bubbles were then adjusted at each scale to reveal 3 cycles per bubble (standard deviations of bubbles were 0.13, 0.27, 0.54, 1.07 and 2.14 degree of visual angle, from fine to coarse scales). The sampled information was then recombined to produce a sparse stimulus (the ‘Final Stimulus’ in Figure 1). The number of stimuli would be further adjusted during the experimental procedure.

In the fMRI experiment, 36 different body action pictures of 12 actors (6 female) expressing 3 expressions (anger, fear, neutral) were also selected from BEAST, including the 12 body images used in the Bubbles experiment. Each whole-body image was split into 3 types of body part (torso with arms, head and legs) and each part was adjusted to the center of an independent image. Whole bodies were also included to be one type of body part, resulting in 144 body images as experimental stimuli in total. Participants viewed the pictures, subtending 6.86° × 5.43° of visual angles, through a mirror mounted on the head coil (mirror size: 3.12 cm × 2.34 cm).

### 2.3. Experimental Tasks

#### 2.3.1. Bubbles Task

Participants performed emotion categorization of a Bubbles stimulus (the ‘Final Stimulus’ in Figure 1) into one of three emotional types by pressing the ‘1’ to ‘3’ key on the upper left side of one computer keyboard. The experiment comprised 4 sessions and 1920 trials in total (160 presentations of each body). Participants viewed each stimulus freely until they pressed one of the buttons. That is, reaction time was not limited, and the next trial would begin after the participant had chosen one emotional type. Participants were required to have a short break after 160 trials. Most of the participants could finish all the trials in 1 h. During the experiment, we collected two datasets of masks: the bubbles masks (each in the second row of Figure 1) (1) led to correct response; (2) led to error response. The datasets were used for analysis after experiments. The sampling density (i.e., the total number of Gaussian bubbles in each bubbles mask) was adjusted on each trial, independently for each expression, to maintain 75% accuracy. Two participants were excluded as they failed to achieve above the accuracy. Experimental programs were performed by scripts [27] on MATLAB platforms (R2017b).

#### 2.3.2. fMRI Behavioral Task

The whole experiment consisted of 4 functional runs and a structural anatomical scan. During the functional runs, participants were required to classify the body stimulus (whole body, torso with arms, heads, legs) into a certain type of bodily expression (anger, fear and neutral). Within each run, there were 108 trials. Procedures were edited and performed through E-Prime 2.0 (Psychology Software Tools, Inc., Pittsburgh, PA, USA). Each functional run started with a white spot presented for 6 s. Within each trial, participants needed to fixate on a white cross at the center of the screen. The fixation duration was chosen from a range of 2–6 s (average: 4 s), which approximates the ISI durations in previous emotion-related tasks [53,54,55]. Then, one stimulus followed and was kept on the screen for 2 s, during which participants were required to indicate its expression type quickly and accurately by pressing one of the three response buttons. The sequence of pressing buttons was counterbalanced across participants. An additional fixation cross was presented for 10 s after 36 trials, in order to increase the fMRI design efficiency (Figure 2).

### 2.4. Data Acquisition

#### 2.4.1. Bubbles Data

To record the body information diagnostic for each expression, the bubbles masks were computed. We added up all the bubbles masks (grayscale of pixels for each scale and each participant) of the datasets which participants correctly responded to. Similarly, the masks that led to error responses were added. To build proportion images, the correct-response mask was divided (independently computed for each pixel) by both the correct-response and error-response mask for each SF band. Then, we employed the Stat4Ci toolbox [56]. These proportion images were smoothed with a Gaussian filter of a standard deviation of 8 pixels by using the function of ‘*SmoothCi*’. After smoothing, these images were transformed into *z* scores using the function of ‘*ZtransSCi**’* to normalize the grayscales of each mask. Diagnostic information was tested on the z-transformed data by applying the cluster test from, using the function of ‘*StatThresh**’* with a significance threshold of *p*-value < 0.05, and cluster *t*-threshold > 2.7.

#### 2.4.2. fMRI Data

The MRI scanning was carried out at the Brain Imaging Research Center of Southwest University, Chongqing, China, using a 3T SIMENS Trio Tim Syngo MR B17 scanner (Siemens Medical, Erlangen, Germany). A gradient-echoplanar imaging (EPI) sequence (scanning parameters: field-of-view/slice thickness: 192/3.5 mm; voxel size: 3.0 × 3.0 × 3.5 mm^3^; matrix: 64 × 64; number of slices: 33; TR/TE: 2000/30 ms; flip angle: 90°) was unified for all runs. Each run consisted of 342 functional volumes. Structural images were acquired through a three-dimensional sagittal T1-weighted magnetization-prepared rapid gradient echo (scanning parameters: field-of-view/slice thickness: 256/1 mm; TR/TE/TI: 1900/2.52/900 ms; voxel size: 1.0 × 1.0 × 1.0 mm^3^; flip angle: 9°; matrix: 256 × 256).

### 2.5. Data Analysis

#### 2.5.1. Bubbles Data

As our aim was to show diagnostic information used in each SF band for a given expression, we summed the number of the significant pixels presented in each SF band and divided that by the sum of all the pixels in the corresponding band. This computation was to reveal the relative use of SF bands across expressions and actors, in terms of the *proportion* of diagnostic information per band [51]. We also summed the different information from 5 SF bands to build the whole diagnostic information for each expression and actor. In order to determine where the diagnostic information could be located, we further divided each body stimulus into three parts: head (including hair and neck), legs (including feet) and torso with two arms and hands. Then, we summed the number of significant pixels presented in each body part of different bands and divided that by the sum of all the pixels in the same part, for each participant, in terms of the *proportion* of diagnostic information per body part. A two-way repeated-measures ANOVA on this diagnostic proportion was conducted with expression (anger, fear, neutral) and body part (head, torso with arms, legs). The Greenhouse–Geisser method was used for sphericity corrections. Post hoc multiple comparisons were conducted on *p*-values with the Bonferroni correction. Effect sizes for each comparison are reported in the Results.

#### 2.5.2. fMRI Behavioral Data

To compare with the statistical results from the Bubbles experiment, we also conducted repeated-measures ANOVAs with Expression (three levels: anger, fear and neutral) and Body (four levels: whole body, torso with arms, legs, and head). We found there might be a response bias that participants preferred to classify the unidentifiable bodies as neutral. Therefore, we adopted an unbiased index *H_u_* [57] which took account of every response bias by multiplying hit rate (ACC). *H_u_* was computed for each condition and subject.

### 2.6. fMRI Localizer and ROI definition

A separate group of 18 participants (10 females, mean age ± s.d. = 22.39 ± 2.06 years) were recruited for one functional localizer task (240 volumes). This localizer experiment consisted of 5 conditions: whole bodies, body parts, hands, tools and chairs [58]. Each condition consisted of 13 different grayscale images (450 × 600 pixels) on a white background. The localizer scan comprised a fully randomized sequence of 25 blocks and ran for 8 min 1 s in total. Scanning started with a white spot presented for 6 s. Within each category block, fixation cross was presented at the center of the screen for 1 s and then 13 different images were randomly presented in the center of the screen. Each image was kept for 800 ms with a blank inter-stimulus interval (ISI) of 200 ms. One image in the stimuli sequence was repeated once. Participants were required (1) to press a button with their right index finger when the repeated image appeared and (2) to pay attention to all 14 images of the stimuli sequence. Interval between two blocks was 4 s. We used this task to identify the selective brain area of perceiving whole bodies and hands for RSA. Scanning parameters were the same as the main fMRI experiment.

Four functional ROIs were defined at the group level of all the participants’ brains: left EBA, right EBA, left FBA and right FBA. The preprocessed data were analyzed using a GLM for each participant. The model included 5 experimental condition regressors and 6 motion correction regressors. In the group-level analysis, four ROIs were selected from the group-level analysis: the EBA and FBA were defined using the t-map of contrast [whole-bodies + body-parts > chairs] [58]. The threshold for ROI definition was set at *q* (FDR) = 0.05. The EBA covered 684 voxels (left) and 835 voxels (right) in the occipito-temporal cortex. The FBA covered 155 voxels (left) and 171 voxels (right) in the fusiform gyrus.

### 2.7. Image Data Preprocessing

Brain imaging data were preprocessed by using the CONN functional connectivity toolbox (version: 16.b; https://www.nitrc.org/projects/conn/; updated on 15 June 2016). First of all, these functional images were slice-timing corrected, realigned and un-warped, and outlier detected (ART-based scrubbing). They were identified as an outlier if (1) the head displacement of any frame surpassed the threshold of 0.9 mm, and (2) the global mean intensity of any frame surpassed 5 standard deviations above the mean intensity of the entire scan. Due to the constraints of head motion and whole-brain intensity, five subjects were excluded. Functional images were co-registered to each subject’s gray matter image segmented from the corresponding high-resolution T1-weighted image, then spatially normalized into a common stereotactic Montreal Neurological Institute (MNI) space and smoothed by an isotropic three-dimensional Gaussian kernel with 6 mm full-width at half-maximum (FWHM).

### 2.8. fMRI Activation Analysis

The whole-brain GLM analysis was performed based on the toolbox of SPM12 software packages (https://www.fil.ion.ucl.ac.uk/spm/; updated on 13 January 2020). The statistical model consisted of 12 regressors of experimental conditions and 6 regressors of head motion parameters. We applied a 3 × 4 ANOVA with emotion and body part to analyze the group random effects. We focused on the main effect of expression conveyed by the whole body or torso with arms. To further test the expression effects, the paired-sample *t*-test was performed on the contrast of ‘anger > neutral’ and ‘fear > neutral’ under both the whole body and torso with arms. The statistical tests were performed at the *q* = 0.05 level corrected for multiple comparisons using the false discovery rate (FDR).

Additionally, in order to evaluate the overlap of activations within the cluster under the contrasts of ‘anger > neutral’ and ‘fear > neutral’ of the whole body and torso with arms, we computed the Sørensen–Dice coefficient (Dice, 1945):R_overlap_ = 2 × V_overlap_/(V1 + V2).(1)

V_overlap_ represents the number of voxels in the common activation region of the two contrasts. V1 and V2 represent the number of voxels in each overlapped cluster.

### 2.9. Constructing Candidate RDMs

We constructed 12 candidate representational dissimilarity matrices (RDMs) to simulate how the ROIs distinguish the bodily conditions and decode their emotional information (Figure 3). The matrices predicted 7 different candidate models: 3 body-effect models (body-separate, body-pattern1 and body-pattern2), 3 emotion-effect models (emotion-separate, emotion-pattern1 and emotion-pattern2) and a random model. Among these models, body-separate, body-pattern1, emotion-separate, emotion-pattern1 were categorical models where two conditions were identical categories (dissimilarity = 0) or different categories (dissimilarity = 1). For body-separate, each bodily condition represented one independent category. For body-pattern1, whole body and torso with arms represented the same category. We designed body-pattern1 by assuming that the ROI representations of the torso–arms and the whole body would represent the same category for processing body parts, if there was no difference in the similarities of the true representations with the body-pattern1 and body-separate model. In other words, body-separate and body-pattern1 together examined whether the EBA extracted diagnostic information mainly from the torso with arms. Similar, the emotion-patterns focused on whether two stimuli shared emotional information. Emotion-separate predicted that each emotional condition represented one independent category. Emotion-pattern1 predicted that the emotion category could be distinguished from the whole body and torso with arms. The 2 models together examined whether the ROIs could abstract the same emotional information from the torso with arms.

The above predictions differentiated whether the ROI representations belonged to independent categories, which might not depict the real similarity relations well. Therefore, we additionally used 2 special candidate models: body-pattern2 and emotion-pattern2. They predicted all the conditions that elicited 5 different prototypical response patterns. We used different ranks, rather than degree, to predict the similarity and all the ranks of each model were rank-transformed from 0–1 before statistical tests. Specifically, in body-pattern2, dissimilarity rank 1 (dark green entries of body-pattern2 in Figure 3) was used to predict the closest category, which might be found in the relationship between the torso with arms and whole body, and between the torso with arms and legs. Rank 2 (red entries of body-pattern2) was used between the legs and whole body, between the head and torso with arms and between the head and legs. Rank 3 (yellow entries) was used between the whole body and head. Based on body-pattern2, emotion-pattern2 distinguished the emotion category from the relationship of the whole body and torso with arms (dissimilarity rank = 0–6). The remaining red and yellow entries of in the emotion-pattern represented dissimilarity ranks 7 and 8, separately.

### 2.10. Representational Similarity Analysis

Representational similarity analysis (RSA) was performed using the scripts according to *rsatoolbox* [59,60]. For each participant, response patterns were extracted from the t-maps of the 12 conditions across voxels inside the 4 ROIs. Then, the true RDM was computed by quantifying the degree of dissimilarity (1 minus correlation) of the response patterns for each pair of conditions, for each of the ROIs. The group RDMs were obtained by averaging all the individual RDMs. In addition, multidimensional scaling (MDS) and hierarchical cluster trees were used to visualize the similarity structure of each group RDM. We performed the RSA to assess (1) whether each candidate RDM was significantly related to the true RDMs and (2) whether there were differences between any two conceptual RDMs in the degree of relatedness to the true RDMs. Kendall’s τ_A_ correlation was used to compute the two relationships. For each candidate RDM and the true RDM, a two-tailed *t*-test was used to assess whether the correlation was significantly against zero over subjects. The statistical threshold for these results was also *q* (FDR) = 0.05 corrected for multiple comparison.

## 3. Results

### 3.1. Bubbles Results

Figure 4 shows the descriptive results of the diagnostic information from the Bubbles analysis. It presents diagnostic information used on one female actor and one male actor for each expression (rows) and each SF band (columns 2–6). The first column presents an integration of the diagnostic information collapsed across the five SF bands. The final column presents a bar graph representing the diagnostic spectrum for each actor (for the results of the other two actors, see Figure A1 and Figure A2).

Repeated-measures ANOVAs on diagnostic proportion showed that the main effect of expression type was significant (*F* (1.4, 42.00) = 5.542, *p* = 0.014, *η2 p* = 0.156) (Figure 5). Main effects of body part were significant (*F* (2, 60) = 17.716, *p* < 0.001, *η2 p* = 0.371). The interaction between them was also significant (*F* (2.45, 73.49) = 4.901, *p* < 0.001, *η2 p* = 0.140). Further simple effect analysis showed that in anger, the torso with arms (M ± SE: 0.151 ± 0.015) was higher than the head (0.077 ± 0.013, *p* = 0.003) and legs (0.050 ± 0.008, *p* < 0.001), while there was no significant difference between the head and legs (*p* = 0.335). In fear, the head (0.214 ± 0.029, *p* = 0.003) and torso (0.191 ± 0.027, *p* < 0.001) were larger than the legs (0.089 ± 0.023), while there was no significant difference between the head and torso (*p* > 0.999). In the neutral condition, there was no significant difference between any pair of the three parts (head: 0.234 ± 0.048, torso: 0.211 ± 0.037, legs: 0.187 ± 0.036; *p* > 0.05).

### 3.2. fMRI Behavioral Performance

Behavioral results (Figure 6) showed significant main effects of expression (*F* (2, 36) = 27.381, *p* < 0.001, *η2 p* = 0.603) and body (*F* (2.244, 40.395) = 236.183, *p* < 0.001, *η2 p* = 0.929). The interaction between expression and body was also significant (*F* (2.984, 53.708) = 42.275, *p* < 0.001, *η2 p* = 0.701). To further examine which body part can be better perceived, simple effects analyses were used to compare the *H_u_* of all the bodies under each expression condition. There were significant differences among body parts for the anger condition (*F* (3, 16) = 238.242, *p* < 0.001, *η2 p* = 0.970). Whole body (0.329 ± 0.015) was performed better than legs (0.043 ± 0.008, *p* < 0.001) and head (0.001 ± 0.001, *p* < 0.001). Torso with arms (0.307 ± 0.016, *p* = 0.015) was also performed better than legs (*p* < 0.001) and head (*p* < 0.001). However, there were no differences between torso with arms and whole body. Legs were performed better than head (*p* < 0.001). For the fear condition, there were significant differences among bodies (*F* (3, 16) = 53.272, *p* < 0.001, *η2 p* = 0.909). Whole body (0.313 ± 0.016) was performed better than torso with arms (0.268 ± 0.019, *p* = 0.029), legs (0.055 ± 0.013, *p* < 0.001) and head (0.030 ± 0.008, *p* < 0.001). Torso with arms was performed better than legs (*p* < 0.001) and head (*p* < 0.001). However, there was no differences between legs and head (*p* = 0.827). For the neutral condition, there were significant differences among bodies (*F* (3, 16) = 11.052, *p* < 0.001, *η2 p* = 0.674). Whole body (0.183 ± 0.009) was performed better than torso with arms (0.129 ± 0.011, *p* = 0.007), legs (0.090 ± 0.008, *p* < 0.001) and head (0.117 ± 0.008, *p* = 0.001). Torso with arms was performed better than legs (*p* < 0.009). Head was also performed better than legs (*p* < 0.045). However, there were no differences between torso with arms and head.

### 3.3. Brain Activations

A whole-brain ANOVA (flexible factorial design) was conducted on the two within-participants factors expression (anger, fear, neutral) and body part (whole body, torso with arms, legs, head) at the group level. The interaction of expression and body part was observed in clusters of the medial frontal cortex, right anterior insula and precuneus. The main effect of expression was found in clusters in the frontal lobe, parietal lobe, fusiform gyrus, post cingulate gyrus, caudate, insula and cerebellum. The main effect of body part was found mainly in the supplementary motor area and occipito-temporal cortex close to the EBA (for more details, see Figure A3 and Figure A4, Table A1). Figure 7 shows the activation difference of the contrasts ‘anger > neutral’ and ‘fear > neutral’ under the condition of whole body and torso with arms. All the contrasts yielded clusters of the left occipito-temporal cortex, showing higher activity for expression-related processing. The whole body and torso with arms showed overlap in 21 voxels (R_overlap_ = 0.50) for the ‘anger > neutral’ contrast and 14 voxels (R_overlap_ = 0.48) for the ‘fear > neutral’ contrast (Figure A5). This indicated that the EBA may contribute to encoding the diagnostic information used in bodily expression perception.

### 3.4. Representations of EBA and FBA

Three main results (to be quantified in subsequent analyses) were found by visual inspection of the four RDMs (Figure 8). (1) All of the RDMs exhibited a dominant body part effect that the whole matrix can be split into sixteen 3 × 3 sub-matrices. Within each sub-matrix, the pairwise correlation tended to be in a similar degree, while it was different between adjacent sub-matrices. (2) The RDMs seemed to contain emotion effects that the neural representations between emotional stimuli shared a low degree of dissimilarity, and neutral stimuli showed large dissimilarity with emotional stimuli. (3) For the EBA RDM, the emotion effects seemed to appear on specific body parts such as whole body and torso with arms, which is consistent with our assumption that torso with arms is the diagnostic body part for emotion recognition. (4) For the FBA RDM, the whole body showed large dissimilarity with all of the three split body parts.

Multi-dimensional scaling (MDS) arrangements and hierarchical plots were performed to visualize the dissimilarity structure arranged by all conditions, and they generally revealed three separate clusters: one for whole body, one for large body parts (torso with arms and legs) and one for head. Torso with arms and legs produced similar responses in both the EBA and FBA. However, the visualizations revealed at least two main different organization of clusters for the EBA and FBA: (1) in the MDS plots, the whole body showed the largest distances from the head for the two EBA ROIs, while the whole body showed large distances from all of the three split body parts for both the FBA ROIs. (2) In the dendrogram, for the EBA, torso with arms and legs were grouped into one cluster, then they were directly grouped with whole bodies. For the FBA, after the three body parts were grouped together into one cluster, whole body was grouped with the cluster. Statistical inference was needed to further examine the relationships.

### 3.5. Statistical Inference of RSA

Statistical inference was performed to test whether each candidate RDM was significantly related to the true RDMs. The relatedness was tested using the signed-rank test. The four bar graphs (Figure 9A) showed that (1) body-pattern1, body-pattern2, body-separate and emotion-pattern2 were positively related to the EBA and FBA RDM except the random, emotion-pattern1 and emotion-separate model. (2) In general, the candidate models predict lower similarities with the FBA, indicating the predictions might not fit the FBA representations well. (3) Among the significant models, emotion-pattern2 and body-pattern2 had high correlations with the true RDM (EBA and FBA, both left and right) while body-separate and body-pattern1 had relatively low correlations, and the results need further pairwise comparisons.

Next, we also tested whether any two candidate RDMs differed in their relatedness to the true RDMs. The upper triangular matrices (Figure 9B) showed that for the correlation to the EBA, (1) emotion-pattern2 was more correlated than body-pattern2 (left EBA: *q*(fdr) < 0.05; right EBA: *q*(fdr) < 0.01). Body-pattern2 was more correlated than body-separate (left and right EBA: *q*(fdr) < 0.01). Among these candidate models, emotion-pattern2 was the best model to simulate the representation of the EBA RDMs. (2) However, there were no significant differences between the correlation of body-separate (to the EBA) and the correlation of body-pattern1 (to the EBA). Both body-pattern1 and body-separate correlated more than the random model (left and right: *q*(fdr) < 0.01). (3) There were no significant differences between the correlation of emotion-pattern1 and the correlation of the random model. Emotion-separate was the worst model to simulate the EBA representation similarities.

Similarly, for the correlation to the FBA, (1) emotion-pattern2 was more correlated than body-pattern2 (left and right FBA: *q*(fdr) < 0.05), and body-pattern2 was more correlated than body-separate (left and right FBA: *q*(fdr) < 0.01). This was consistent with the correlation to the EBA. Emotion-pattern2 was also the best candidate RDM to simulate the representation of the FBA RDMs. (2) However, body-separate was also more correlated than body-pattern1 (left and right FBA: *q*(fdr) < 0.05). Body-pattern1 correlated more than emotion-pattern1 (left and right FBA: *q*(fdr) < 0.01). (3) There were no differences between the correlation of emotion-pattern1 and the correlation of the random model to the left FBA (*q*(fdr) > 0.05), or between the correlation of the random model and the correlation of emotion-separate (left and right FBA: *q*(fdr) > 0.05). Body-pattern1 was more correlated than the random model (*q*(fdr) < 0.05).

## 4. Discussion

The current study explored the diagnostic parts for bodily expression recognition and their mechanisms by analyzing the brain representations of bodily expression. To illustrate, clenched fists and flexed arms of hit gestures reliably allowed observers to categorize the emotion as anger. Similarly, a backward-leaning torso, arms in front of torso and hands shielding the body all reliably indicated fear. Our findings were generally consistent with behavior types used for expression decoding and encoding in previous expression communication studies [10,18,61]. Moreover, in the fMRI experiment we found that the response patterns in the EBA carried information to clearly distinguish different body parts. In contrast, the FBA only distinguished between the whole body and body parts. Furthermore, the EBA decoded the information which may be used for further expression perception.

### 4.1. Torso with Arms as Diagnostic Body Parts

Previous researchers manipulated behavioral type (spatial form, such as head tilted up or down) or quality (spatiotemporal properties, such as speed and energy) to study bodily behaviors, and identified some specific behaviors which can be used to perceive both static and dynamic expressions [17,62]. For example, hot anger (or rage) portrayals can be expressed by a forward lean or movement [5]. However, in more cases, bodily expression is transmitted through flexible and variable motion patterns [10], and it is difficult to focus on all the patterns in one experiment. We therefore adopted another perspective which directly focused on the diagnostic body parts.

The main contribution of the Bubbles experiment is that we identified the diagnostic body parts to summarize the diagnostic features, for understanding the mechanisms underlying expression perception. Here, the Bubbles methods we used were not entirely consistent with previous studies, which revealed the diagnostic information in different SF bandwidths [51,52]. Instead, we additionally divided the whole body into three body parts, and analyzed the proportions of diagnostic information for each body part. The purpose of doing this analysis is that we contrived to integrate the flexible postures or movements. Previous studies used the Bubbles paradigms on facial expression, however, facial expression and bodily expressions differed a great deal in their structures or configurations [62,63]. Bodily expressions have flexible postures or movements, therefore, there were many limitations in generalizing the results of diagnostic features to other bodily behavior patterns even in perceiving the same expression. However, at the level of body parts, there were significant differences in the amount of diagnostic information that the body parts contained. If two irregular dynamics share ‘similar’ posture or movement, they probably share the ‘same’ diagnostic body parts for perception. That is, to distinguish whether the bodily expression is anger or fear, we can simply pay attention to the torso with arms.

The current experiment showed two characteristics consistent with previous studies. (1) Diagnostic feature selection basically depends on the information structure provided by visual inputs [13]. For perceiving anger, observers selected the area of the thighs as a diagnostic feature for one of the female actors, however little information was selected from the thighs for one of the male actors (Figure 4). For the male actor, observers considered visual information only from his fist. This variation in diagnostic features cannot be attributed solely to perceived gender association with certain expressions, because the two actors differed subtly in body postures. (2) Diagnostic feature selection also depends on flexibly utilizing the spatial locations of body parts. For example, extracting head information differed obviously between the Bubbles task and fMRI experiment. The diagnostic features for heads originally contained both the neck and lower half of the head, while they could not be recognized in fMRI experiments. We inferred the reason was that head orientation was difficult to be identified when heads were presented alone, but may be easy when integrated with torso parts. This is consistent with the notion that spatial location was extracted by a top-down processing mechanism [13,64], which is modulated by task requirements, memory representations and strategies [13,52,64]. Here, the absence of spatial location for head orientation leads to more flexible strategies in utilizing memory representations of diagnostic features.

### 4.2. Brain Activations Related to Body Parts

Brain activation results were generally consistent with the body recognition literatures [28,46,58,65]. Notably, the angry vs. neutral contrast of leg stimuli revealed stronger activations of the lingual gyrus, the inferior parietal lobule (IPL), supplementary motor area (SMA), thalamus and the anterior insula (AI). The lingual gyrus has been demonstrated to be involved in processing both human faces [66] and bodies [67]. The lingual gyrus was activated during passive viewing of body parts [67]. Therefore, the stronger activation of angry torsos relative to neutral torsos possibly reflected processing information of body motion.

The IPL also modulates the perception of facial expressions and interpretation of character information [68]. Importantly, the IPL plays a causal role in processing fearful bodies [69]. However, we only observed IPL activation during the ‘anger versus neutral’ contrast. More experiments are needed to explain this finding. Activation of the SMA upon viewing angry legs implies that the area is collecting information on body motion [70,71], given that the region is involved in planning or preparation of movements [72]. Our results also found the activation of the caudate in the thalamus was enhanced in this contrast. Studies found that the thalamus and AI could be linked with the experience of unpleasant and highly arousing affect [38]. In particular, the AI may be responsible for the integration of interoceptive awareness with feelings of disgust and phobia subjects with higher interoceptive awareness could require less AI activity to maintain similar behavior when viewing phobia stimuli [73,74]. This function may explain why AI activation occurred when viewing nondiagnostic body parts (e.g., head, legs). However, the AI is critical to the salience network, particularly in switching access to working memory and attentional resources to detecting salient events [74].

Other subcortical structures, such as the amygdala, were not found in the anger vs. neutral contrast, nor in other contrasts. We could not draw conclusions that these regions were not involved in the perception of bodily expression, because the results might be dependent on the experimental design. As we mentioned in the Introduction, the current design may not evoke a rapid response of the amygdala.

### 4.3. Neural Representation of Diagnostic Body Parts

We confirmed the significance of the torso part for expression perception by examining the brain activities of separate body parts. Regions activated by the torso part and whole body overlapped a great deal in the left occipito-temporal lobes near the EBA which is sensitive to body parts and biological motion [37,46,75]. This indicated that the EBA might process the bodily expression by extracting the information from the torso and arms. Therefore, we further examined the dissimilarity structures of neural representation in the EBA. First of all, we found that the EBA RDMs were more correlated to body-pattern2 than body-separate and body-pattern1. There were no differences in the correlation between any pair of the latter two models with the EBA RDMs, indicating that among the three body parts, the neural representation for the torso part was most similar to that of the whole body. As the FBA differentiates body configurations [75,76], the same analysis was also applied to the neural representations in the FBA. We also found the effects of body-pattern2, however, the FBA RDMs were more correlated to body-separate than body-pattern1. This indicated that the FBA preferred to decode the body parts as independent categories. Poyo Solanas and colleagues [76] demonstrated that both the EBA and FBA could decode the information of limb contraction. However, their neural representations differed. Our results may contribute to showing the representational difference between the EBA and FBA.

Second, we showed an organizational structure for representations in the EBA and FBA, related to distinguishing posture or movement of body parts. This is consistent with previous studies [43,77] that showed that the EBA could produce response to body parts at the semantic level rather than at the physical property level. However, these researchers did not clarify the posture or movement of body parts. Body-separate was better than body-pattern1 in the correlation to both sides of the FBA. This reflected that the FBA might not be as sensitive as the EBA to decode the torso with arms. The FBA was therefore not a good brain area sensitive to the diagnostic body parts. This functional difference is consistent with the studies emphasizing the functional specialization of the EBA and FBA [76,78].

Furthermore, postural features from the torso and arms possibly drive the bodily expression perception. For example, limb contraction drives fear perception [76]. Our results further indicated that decoding the posture was probably derived from diagnostic body parts. The EBA RDMs were more correlated to emotion-pattern2 than body-pattern2. This provided evidence that the torso with arms was decoded in a similar way as the whole body. Combining the facts that the EBA plays a role in action perception [12] and is connected to parietal cortex regions [78], we inferred that the EBA possibly transmits the movement information of the upper limbs for more abstract perception. Taken together, the main contribution of the current experiment was that the EBA may convey diagnostic information for perceiving bodily expressions.

### 4.4. Limitations and Future Expectations

We employed methods that have been previously used to investigate diagnostic information and neural representations. However, the approaches we used have limitations in some respects: (1) group-level functional ROIs were adopted, rather than individual-subject level ones, limiting the accuracy of the functional localization of MVPA effects. (2) Only a limited set of basic emotions (fear and anger) were employed, limiting the generality of the conclusions. (3) Likewise, only static stimuli were used—a different answer might be obtained for information conveyed by body motion. For instance, recent works on the ‘forrest dataset’ [79] investigated emotional representations during movie watching [80,81]. Dynamic bodily expressions unfolded time-varying and complex emotions, which were closer to real-life experiences. Dynamic features conveyed more details, such as the time to rise, or the probability of resurgence [82]. Future research should expand the types of emotion and stimuli to further investigate the features and related neural mechanisms of bodily emotion.

## 5. Conclusions

Behavioral evidence supports that diagnostic emotional information is involved in the torso (including arms and hands) for perceiving both anger and fear. The body part and whole body also share similar neural representations in the EBA. Furthermore, we demonstrated that the canonical area of the EBA distinguished the posture or movement for decoding bodily emotion. In sum, both behavioral and imaging results showed that the action or postures of upper body parts can provide core features for emotion recognition.

## Figures and Tables

**Figure 1 brainsci-12-00466-f001:**
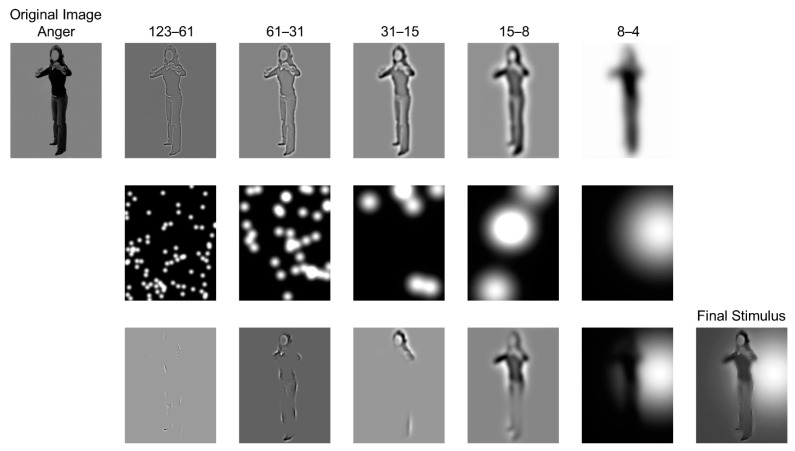
Illustration of generating Bubbles stimulus. As shown in the first row, each original body stimulus was decomposed into 5 scales of five spatial frequency bandwidths (123 to 4 cpi). Then, in the second row, each bandwidth was independently and randomly placed Gaussian window bubbles. The third row shows the body information revealed by bubbles of each scale and the sum of information across scales. The final stimulus summed the 5 leftmost pictures on the row, and it was then applied in the formal experiment.

**Figure 2 brainsci-12-00466-f002:**
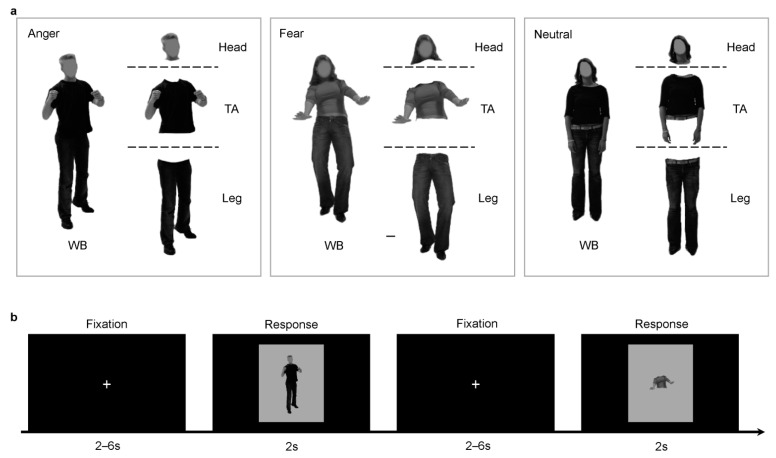
(**a**). Stimuli illustration of the 3 bodily expressions and 4 body parts. (**b**). fMRI task procedures.

**Figure 3 brainsci-12-00466-f003:**
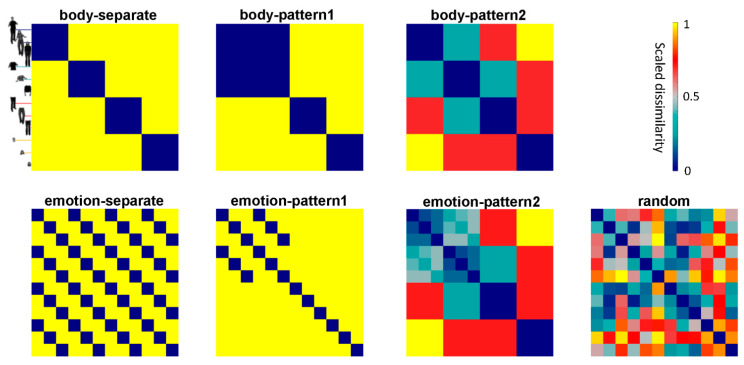
Candidate models. Body-separate, body-pattern1, emotion-separate and emotion-pattern1 are categorical models of simulating the similarity of the BOLD activation patterns induced by the emotional categorization task, if the body or emotion factors independently dominate the underlying representations. Body-pattern2 assumes that the similarities of activation patterns induced by torso + arms, legs and head with that induced by whole body vary from high to low. Emotion-pattern2 combines emotion-pattern1 and body-pattern2, assuming that torso + arms and whole body share similar patterns for emotion categorization.

**Figure 4 brainsci-12-00466-f004:**
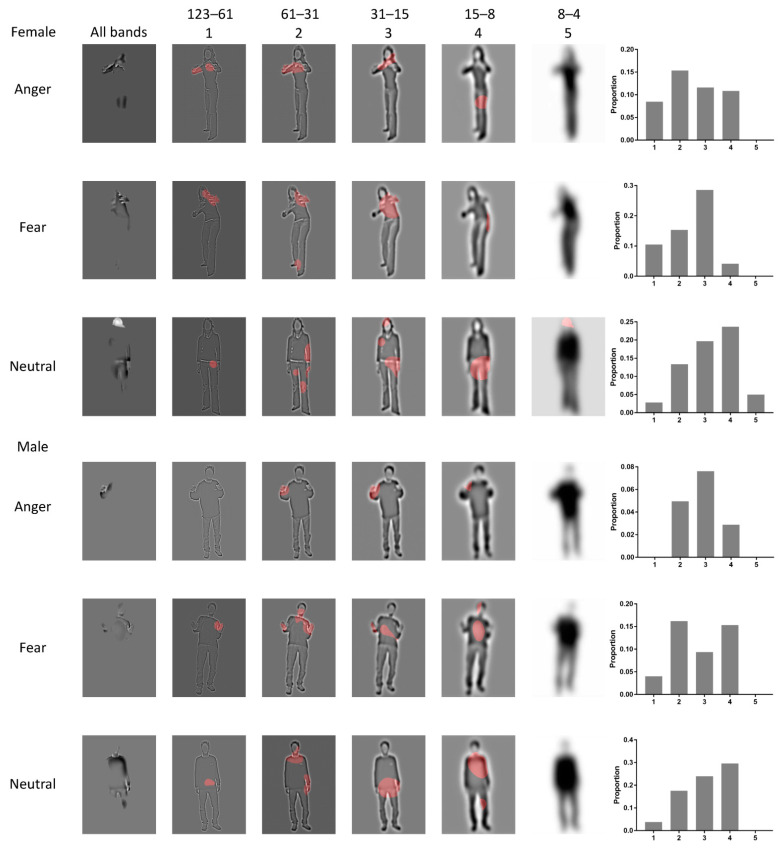
Diagnostic information revealed by the Bubbles experiment. The significant body information (red regions) for categorizing each bodily expression is displayed in a separate row. The first three rows show the three expressions by a female actor and the latter three rows those by a male actor. The first column shows the diagnostic SF features overlaying all the SF bands sampled in our experiment. The next five columns show the SF features of each band, respectively. The last bar graph is about the diagnostic SF spectrum for each expression (proportion of the diagnostic information per band). The numbers at the top show the range of each bandwidth (unit: cpi). The numbers at the top correspond with those below each bar graph.

**Figure 5 brainsci-12-00466-f005:**
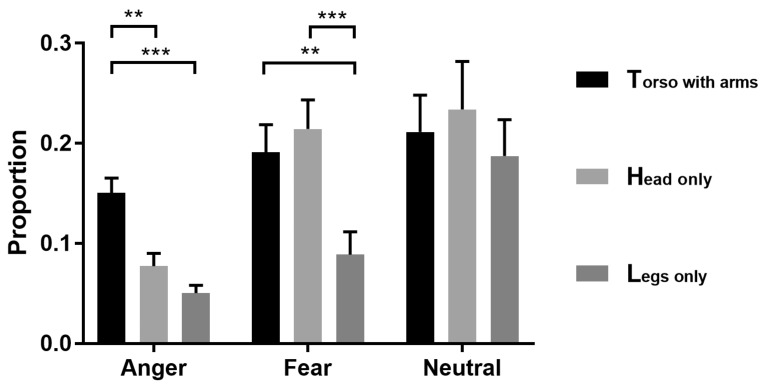
Bar graph for results of the Bubbles experiment. Each bar represents the diagnostic pixel proportion (mean + s.e.m.) in the body parts of torso with arms, legs and head for classification as anger, fear and neutral. ** *p* < 0.005; *** *p* < 0.001.

**Figure 6 brainsci-12-00466-f006:**
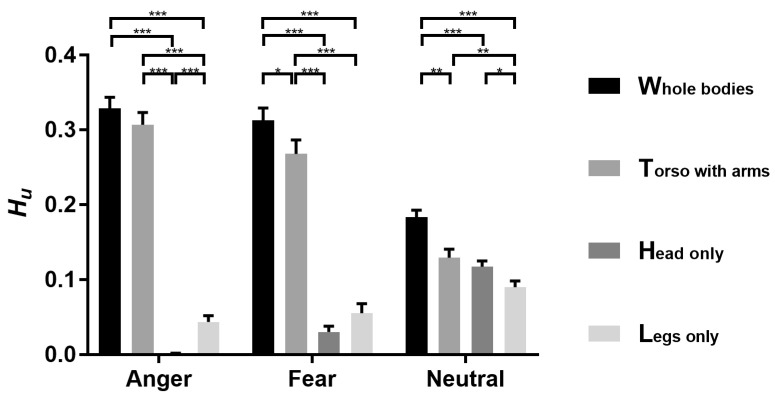
Bar graph for behavioral results of fMRI experiment. Each bar represents the behavioral performance (*H_u_*, see the main text; mean + s.e.m.) for classifying the WB, TA, legs and head into anger, fear and neutral, respectively. * *p* < 0.05; ** *p* < 0.01; *** *p* < 0.001.

**Figure 7 brainsci-12-00466-f007:**
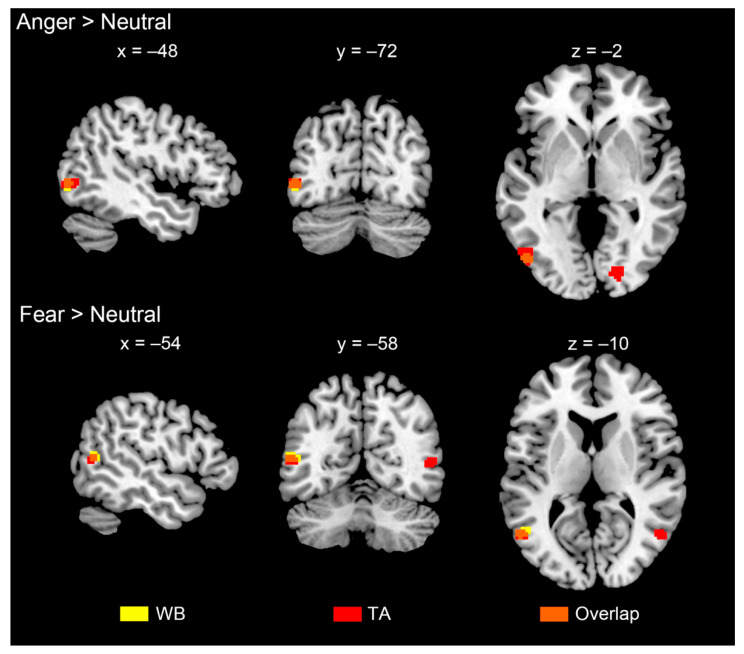
Group analysis results for the contrast of ‘anger vs. neutral’ and ‘fear vs. neutral’ under whole body (WB, yellow clusters) and torso with arms (TA, red clusters) conditions. WB and TA were overlapped in orange clusters. The clusters were significantly located in occipitotemporal cortex around EBA.

**Figure 8 brainsci-12-00466-f008:**
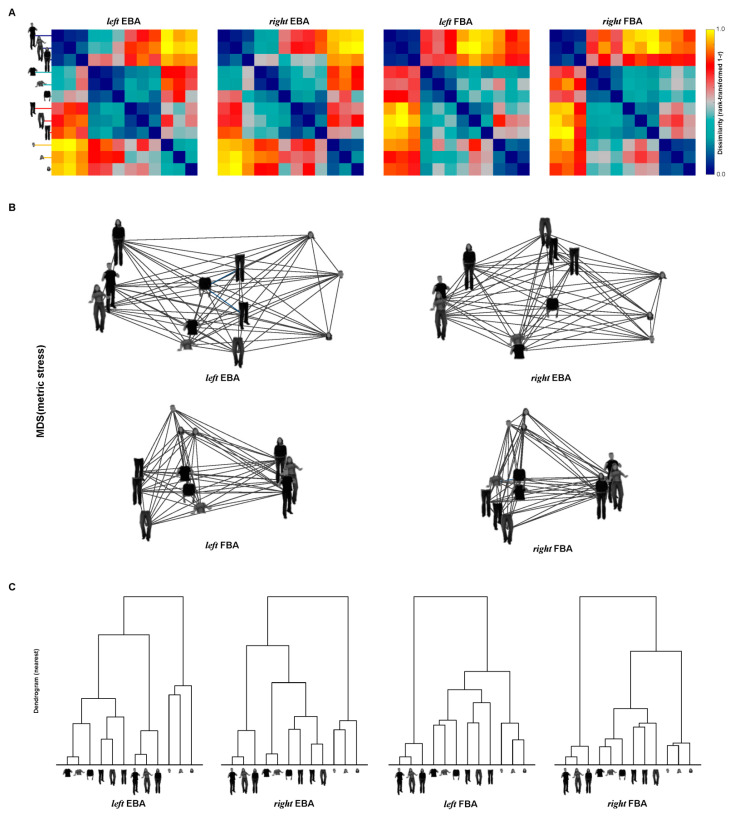
Representation structures in EBA and FBA. (**A**) True RDMs, averaged across subjects for the four ROIs, show the neural dissimilarity (1 – r) between any two of the body parts. (**B**) MDS, calculated based on the RDM matrices, plotting the pairwise distance in a 2D space. The distances reflect the response-pattern similarity: the pairs which are located next to each other shared similar response patterns, while those far away from each other had dissimilar response patterns. (**C**) Dendrogram, grouping the body parts (nearest neighbor), aiming at revealing their categorical divisions.

**Figure 9 brainsci-12-00466-f009:**
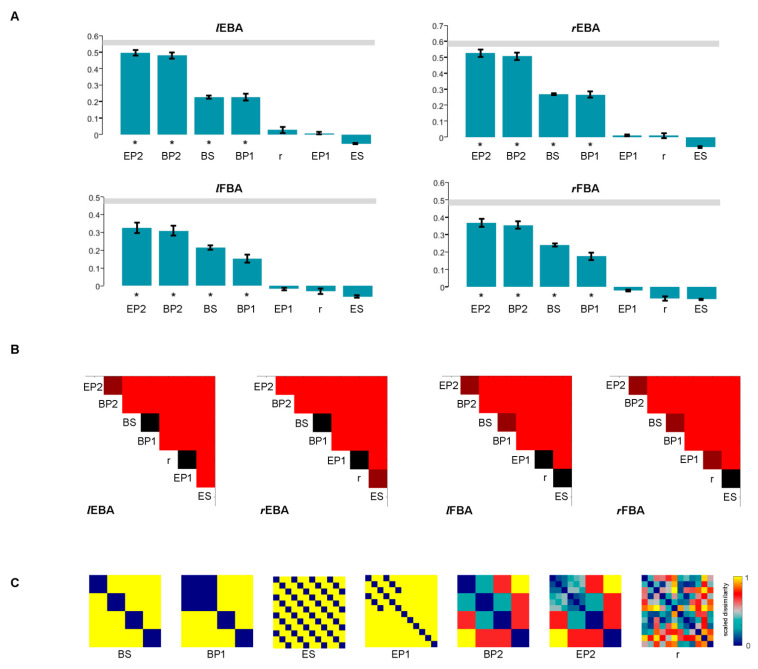
Statistical test results. (**A**) Correlation (Kendall’s rank correlation coefficient τ_A_) between the true RDMs and the candidate RDMs, respectively. The correlation coefficients were tested using a default one-sided signed-rank test. Significant results are marked by one ‘*’ below the bars. (**B**) The difference between any two candidate RDMs in their relatedness to the true RDMs. Each entry represents the significance of the difference tested by a two-sided signed-rank test. The colors of each entry represent different significant thresholds: *q*(FDR) = 0.05 (deep red) and *q*(FDR) = 0.01 (red); the nonsignificant entries are black. (**C**) Candidate models. BS: body-separate; BP: body-pattern; ES: emotion-separate; EP: emotion-pattern; r: random.

## Data Availability

The data and code that support the findings of this study are available from the corresponding author upon reasonable request by e-mail. The data are not publicly available as new data such as the fMRI data under more emotion conditions were created and will be further analyzed in future studies. We will provide the data, code and results of the Bubbles experiments and the task-fMRI experiments, if editors, reviewers and any readers need them to perform validation or any other analysis.

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
