# Peer review of "Features and Extra-Striate Body Area Representations of Diagnostic Body Parts in Anger and Fear Perception"

_brainsci, 2022, doi:10.3390/brainsci12040466_

Round 1
Reviewer 1 Report
Ren and colleagues conducted behavioral and fMRI studies investigating the role of bodily parts in the perception of negative emotions and how these are represented in EBA. The study seems to provide interesting insight in the field of the social and affective neuroscience.
However, few concerns can be raised in this regard. Overall, authors should thoroughly check the manuscript for typos and to improve the clarity of sentences.
- The introduction can be expanded including relevant references on the Bubbles paradigm and how emotional expressions are expressed in the body (e.g., works from Rachel Jack, Lauri Nummenmaa)
- It is not clear whether the pictures included in the behavioral experiment were also in the fMRI one. If not, the authors should provide an explanation behind this, as it would be reasonable to have an overlap between stimuli.
- Further details on the acquisition setup and paradigm should be provided for the behavioral experiment.
- Which was the accuracy threshold the authors employed in the behavioral experiment? This should be clear.
- Figure 3 is not completely clear, I think that some matrices are actually not useful for the understanding of the study, as they do not have much variability. I would suggest to rimodulate the figure to make it more clear in the message for the reader.
- The same can be said for Figure 8, panel B. As they are now, the graphs are not very clear, so they might need to be enlarged or rearranged as they are not very visible in the present form.
- The discussion of the paper might benefit from an additional part, where more naturalistic stimulation is taken into consideration. For instance, there has been recent work on the studyforrest dataset (Hanke et al., 2016) that investigated emotional representation during movie watching (Lettieri et al., 2019, 2021). I think possible limitations due to the use of static stimuli compared to more naturalistic ones should be discussed.
Reviewer 2 Report
The manuscript focuses on specific body parts and their movements for the detection of emotions. Typically studies analyze overall body expression and behavior for feature extraction. This research, on the other hand, used the torso, arms, head motion, and the combination of these body parts and their contribution to emotion detection features. The experiments were conducted using the bubble paradigm and were followed up with functional magnetic resonance imaging experiments. The results showed higher activation in the extra-striate body area as a response to anger and fear. Overall the paper is very well organized with sufficient details in the literature study, experimental design, and analysis of the results. The researchers have also identified the limitations of the study (fewer basic emotions, smaller participant demographic) and stimuli. The results have been thoroughly discussed in terms of fMRI performance, bubble results, and brain stimulation readings. The manuscript is quite informative and well written and the results are backed by sound statistical analysis.
Reviewer 3 Report
The paper is interesting and I generally appreciate it. However, I find a structural limitation in analysing only some ROI in fMRI study omitting all the subcortical structures associated with emotions. Furthermore, the methods are difficult to follow and the motivation for multiple steps in the analyses are not explained.
other points
Line 40. What do the authors mean with the expression “diagnostic body parts”?
Lines 74-80. The introduction of the Bubble paradigm is confusing (e.g decoding visual information by avoiding self-reports which instead used and analysed……) and it is not totally clear the reasons of choice of this specific paradigm
In general in the introduction it is not clear if authors are referring to still or moving body parts
Lines 91-92: studies on patients indicate dissociations between identification of body form and body motion see Urgesi et al., 2006 and Moro et al., 2008. I agree that Eba participates in both the processes, but a reference to the role of the frontal system in action discrimination would be useful.
In Introduction the authors do not consider that angry and fear expressions are associated not only with motion, but also with network of emotions. Although I agree that emotions are mainly transmitted via motion, neural correlates of the two processes are not totally overlapping. I see that the aim of the study is identifying the motion (or maybe postural) information useful to understand emotions but a reference to differences should be put. For example, subcortical emotional-related networks should be introduced (e.g. Tamietto et al., 2007; Van der Stock et al., 2011; Moro et al., 2012).
In figure 1 I cannot understand if the study uses all the stimulus in the bottom line (maybe the sum of stimulus and bubble for each SF?) or only the final one indicated on the right of the figure
Lines 297-298 The torso+arms stimuli make it difficult to understand the specific role of torso and arms separately…. If we assume that torso+arms and the whole body are the same, is an expected result a major role of these body parts together in emotion discrimination?
Figure 3 some indications on how to read the pictures of single models would be useful for not experts in this method
Results: do the authors have some hypotheses on the lack of results in the subcortical structures of the limbic system?
Round 2
Reviewer 3 Report
The authors have responded to my comments and the paper is now ready for pubblication